# Cross-sectional association of *Toxoplasma gondii* exposure with BMI and diet in US adults

**Joel Cuffey** [1]*, **Christopher A. Lepczyk** [2], **Shuoli Zhao** [3], **Nicholas M. Fountain-Jones** [4]

**1** Department of Agricultural Economics and Rural Sociology, Auburn University, Auburn, Alabama, United States of America, **2** School of Forestry and Wildlife Sciences, Auburn University, Auburn, Alabama, United States of America, **3** Department of Agricultural Economics, University of Kentucky, Lexington, Kentucky, United States of America, **4** School of Natural Sciences, University of Tasmania, Hobart, Australia

* cuffey@auburn.edu

**Data Availability Statement:** All Stata-format data files and Stata syntax files necessary to replicate the results are available from the AUrora repository at http://dx.doi.org/10.35099/aurora-80.

**Funding:** JC acknowledges support from the Alabama Agricultural Experiment Station and the

## Abstract

*Toxoplasmosis gondii* exposure has been linked to increased impulsivity and risky behaviors, which has implications for eating behavior. Impulsivity and risk tolerance is known to be related with worse diets and a higher chance of obesity. There is little known, however, about the independent link between *Toxoplasma gondii* (*T. gondii*) exposure and diet-related outcomes. Using linear and quantile regression, we estimated the relationship between *T. gondii* exposure and BMI, total energy intake (kcal), and diet quality as measured by the Health Eating Index-2015 (HEI) among 9,853 adults from the 2009–2014 National Health and Nutrition Examination Survey. Previous studies have shown different behavioral responses to *T. gondii* infection among males and females, and socioeconomic factors are also likely to be important as both *T. gondii* and poor diet are more prevalent among U.S. populations in poverty. We therefore measured the associations between *T. gondii* and diet-related outcomes separately for men and women and for respondents in poverty. Among females <200% of the federal poverty level *Toxoplasmosis gondii* exposure was associated with a higher BMI by 2.0 units (95% CI [0.22, 3.83]) at median BMI and a lower HEI by 5.05 units (95% CI [-7.87, -2.24]) at the 25th percentile of HEI. Stronger associations were found at higher levels of BMI and worse diet quality among females. No associations were found among males. Through a detailed investigation of mechanisms, we were able to rule out *T. gondii* exposure from cat ownership, differing amounts of meat, and drinking water source as potential confounding factors; environmental exposure to *T. gondii* as well as changes in human behavior due to parasitic infection remain primary mechanisms.

## Author summary

*Toxoplasmosis gondii* (*T.* gondii) is a parasite that infects over 10 percent of the US population. *T. gondii* infection can cause serious health problems for some people, but most infections remain undiagnosed and subclinical. When an individual is infected, *T. gondii* can chronically reside in muscle and central nervous system (including brain) tissue. Previous studies have found that individuals with prior exposure to *T. gondii* may engage in

Hatch program of the National Institute of Food and Agriculture, US Department of Agriculture (accession number 1023614). The funders had no role in the study design, data collection and analysis, decision to publish, or manuscript preparation.

**Competing interests:** The authors have declared that no competing interests exist.

more risky and impulsive behaviors, and risk tolerance and impulsivity may be related with individual's diet. Our study examines whether individuals with *T. gondii* exposure have higher body mass index (BMI) and worse diets. We further discuss and test for alternative explanations that prevent us from establishing a causal relationship between *T. gondii* and BMI/diet. Overall, our results show that *T. gondii* exposure is related with higher BMI and worse diets among lower-income females in the US. Our results uncover a novel correlate of BMI and diets, and suggest the importance of investigating the broader public health impacts of chronic *T. gondii* infection.

## Introduction

The protozoon parasite *Toxoplasma gondii* infects over 10 percent of the US population [1], with low-income populations bearing the greatest burden [2]. Most infections remain undiagnosed and subclinical, but *T. gondii* can cause serious health problems for some individuals, including infected fetuses and infants [3]. Subclinical chronic *T. gondii* infection can cause changes in the brain [4], and has been linked to more subtle behavioral changes in humans. *T. gondii* exposure may decrease cognitive function [5] and may increase the risk of mental health problems such as schizophrenia or psychosis [6–7]. Adults with *T. gondii* exposure are more likely to engage in risky behavior such as alcohol consumption [8], risky driving [9], and entrepreneurial activities [10] and they may also exhibit higher levels of aggression and impulsivity [11]. Many of these behavioral changes are also known to increase the chance of having poor diet and being obese. In particular, individuals with greater tolerance for risk or who are more impulsive have higher body mass index (BMI) and worse diets [12–17].

Despite these behavioral linkages, little is known about the role of *T. gondii* exposure in explaining diet or obesity. *T. gondii* has been shown to cause short-term weight gain in rats [18], and is associated with obesity in German populations [19–20] as well as with type 2 diabetes [21]. However, the relationship with obesity was not found in a sample of Mexican individuals [22], and no link has been found between toxoplasmosis and fatty liver disease [23]. No studies to date have examined whether *T. gondii* exposure can explain differences in overall diet quality.

Our primary aim in this study was to evaluate whether *T. gondii* exposure explains body mass index (BMI), total energy consumption, and diet quality in a nationally-representative sample from the United States. Since the available data do not allow us to establish causality, our secondary aim was to investigate potential mechanisms for any relationship between *T. gondii* exposure and these outcomes.

## Methods

### Data

We used three waves (2009–2010, 2011–2012, and 2013–2014) of the continuous National Health and Nutrition Examination Survey (NHANES) to examine the relationships between *T. gondii* exposure, BMI, energy consumed, and diet quality. The NHANES samples U.S. residents and conducts a detailed survey that contains clinical examinations and laboratory tests. From the 2009–2010, 2011–2012, and 2013–2014 waves, a subset of the respondent blood samples was tested for *T. gondii* IgG antibodies using an enzyme immunoassay. The subset of respondents consisted of all those with surplus serum left after conducting the other regular NHANES laboratory tests, with the exception of pregnant women [1]. We defined *T. gondii*

exposure (i.e. seropositivity) as having *T. gondii* IgG antibody concentration of ≥33 IU/mL; any amount < 33 IU/mL is classified as no exposure (i.e. seronegative) [1]. We also controlled for poverty in our models as respondents under the poverty level were less likely to have serum tested for *T. gondii* IgG antibodies. Furthermore, adjusting the NHANES sampling weights to account for income differences leads to the same results on *T. gondii* exposure prevalence and risk factors as using the original NHANES sampling weights [1]. We therefore used the original NHANES sampling weights in our analyses.

As part of the NHANES clinical examination, respondents' weight (kg) and height (cm) are measured, and BMI is calculated for each respondent. The clinical examination also includes a 24-hour dietary recall component [24]. NHANES aggregates the foods consumed for each respondent and releases data on total energy consumed (kcal). This allowed us to measure the quantity that a respondent eats. In addition, the nutrient consumption data allow us to examine the quality of the diet. Specifically, the Healthy Eating Index (HEI) evaluates 13 different dietary components and measures on a scale of 0–100 how closely a particular diet adheres to the Dietary Guidelines for Americans. Of the 13 dietary components used to score the HEI, greater amounts of total fruits, whole fruits, total vegetables, green vegetables or beans, whole grains, dairy, total protein, seafood or plant proteins, and fatty acids increase the HEI score. Greater amounts of four components decrease the HEI score: refined grains, sodium, added sugars, and saturated fats. A score of 100 indicates the best diet possible. The HEI has been revised twice since it was first developed in 1995; we use the HEI-2015 [25].

We restricted our sample to adults (18+ years old) whose blood samples were tested for *T. gondii* antibodies and who have available BMI and dietary data. Previous research has found much higher *T. gondii* seroprevalence among foreign-born NHANES respondents [1], so we excluded respondents who were not born in the U.S., resulting in a total sample size of 9,853 individuals.

The NHANES waves used in this study provide the only publicly accessible measures of *T. gondii* exposure linked to detailed food intake data for the U.S. population. The NHANES dietary recall data, however, have two limitations relevant to examining diet and *T. gondii* exposure. First, respondents regularly misreport food consumption in self-reported dietary recalls such as the NHANES [26–28] and misreporting is greater for obese respondents [29]. We note though that the dietary recall elicitation method used in the NHANES suffers from substantially less misreporting than other methods [30] and that overall energy intake is underreported by 11% by the method used by NHANES [29]. The second limitation inherent in the NHANES data is the inability to observe food safety and hygiene practices, which may be important in *T. gondii* transmission [31].

To accomplish our secondary aim of examining potential mechanisms and confounders for our results, we modified our sample and data from the main sample described above. The rationale for each modification is given in the Results section below. One subsample consisted of respondents with valid information on infection with *Toxocara canis/cati*. This information was available in 2011–2012 and 2013–2014 NHANES waves. Similar to the *T. gondii* laboratory samples, a subsample of survey respondents' blood specimens were tested for *Toxocara* antibodies. Samples were classified as positive for *Toxocara* if the mean fluorescence intensity was greater than 23.1 and negative otherwise [32]. A total of 6,237 respondents had available *Toxocara* information and also met our sample restrictions above. A second subsample excluded an additional 1,544 individuals who reported participating in vigorous recreational activities at least three days per week, resulting in a sample size of 8,309. A third subsample included only respondents with valid percent body fat information. NHANES used dual-energy X-ray absorptiometry to measure body composition in the 2011–2012 and 2013–2014 waves. A total

of 3,668 respondents meeting our sample restrictions also had valid information on the percent body fat.

A final sample we used to investigate mechanisms consists of respondents in the 2005–2006 NHANES wave. The NHANES waves with *T. gondii* data described above did not have information on cat ownership, though the 2005–2006 NHANES wave included this information. The 2005–2006 wave captured cat ownership through two questions asking respondents whether a cat lives in their house now, and whether a cat lived at their house at some point in the past 12 months. We combined these two questions and measured cat ownership to be whether an individual either has a cat in the house now or had a cat in the house in the past 12 months. We made the same sample restrictions as our main analysis (U.S.-born adults with valid dietary data), yielding a sample size of 3,545 and 1,395 low-income individuals. We note that though this sample does not have available *T. gondii* information it does provide the same outcome measures as our primary sample (BMI, total energy consumption, HEI), allowing us to measure the association between cat ownership and diet-related outcomes.

## Statistical analysis

We used two analytical methods to examine the relationship between *T. gondii* exposure and our outcomes of interest. First, we used linear regression of an indicator for *T. gondii* exposure on continuous BMI, energy consumption, and HEI. We also included an interaction between *T. gondii* exposure and whether the respondent's income is over 200% of the federal poverty level. In these regressions, we controlled for whether income is over 200% of poverty, gender, race/ethnicity (non-Hispanic white [reference category], non-Hispanic black, Hispanic, other race), household size, household size squared, age, age squared, whether married, education (less than high school [reference category], high school graduate, some college, college graduate), and whether the interview was conducted between November and April of the particular calendar year (relative to interviews conducted between May-October). We controlled for the part of the year the interview was conducted (November April vs. May-October) to account for possible seasonal differences in diet behavior. To account for differential energy needs we also controlled for the respondent's BMI in analyses with energy consumption and HEI outcomes. Prior studies found differences between males and females in relationships of behavior with *T. gondii* exposure [33–34], so we estimated separate regression models for males and females. This decision was made prior to data analysis and was not impacted by the results. Linear regressions used the NHANES examination sampling weight and standard errors take into account the NHANES complex sample design.

Because linear regression results may mask varied relationships between *T. gondii* exposure and outcomes at different points in the outcome's distribution [35–36], we estimated a second series of models using quantile regression (QR). QR allows for evaluating relationships between independent variables and the outcome at different quantiles of the outcome's distribution [37]. We estimated QR models with identical outcome and control variables as our linear regression models. We used the NHANES examination sampling weight in our QR analysis and bootstrapped the standard errors 500 times to account for the complex sample design [38]. QR results are presented in the figures for every fifth percentile from the 5th to the 95th percentile of the outcome distribution. Figures display the coefficient on the indicator for *T. gondii* exposure and its corresponding 95% confidence interval. Separate figures display the coefficient on the interaction between *T. gondii* exposure and whether the respondent's income is >200% of the federal poverty level. The coefficient on the indicator for *T. gondii* exposure therefore measures the relationship for respondents <200% of poverty and the interaction measures the difference between respondents <200% of poverty and those >200% of poverty. For all modeling approaches we considered $p \leq 0.05$ as significant.

We used additional statistical analysis to investigate mechanisms underlying our main results. Rationales are described below. In addition to the control variables above, we separately added the following controls: total protein consumption (grams), pork consumption (kcal) as a percent of total energy consumption (kcal), whether the respondent owns a cat, whether the respondent's main source of tap water is a well (vs. a municipal source), and whether the respondent is *Toxocara* seropositive. We also separately examine the association between *T. gondii* exposure and the following additional outcomes: total protein consumption (grams), pork consumption (kcal) as a percent of total energy consumption (kcal), waist-to-height ratio (WHtR), percent body fat. Since 95.6% of our sample did not eat pork in the 24 hours covered by the dietary recall, quantile regression on the percent of total consumption from pork was unstable. We therefore measured this association using linear regression on mean pork percent as well as indicators for being in different parts of the pork consumption distribution [39].

## Results and discussion

### *T. gondii* and outcomes among females and males

Among females <200% of poverty *T. gondii* exposure was associated with an increased BMI at higher levels of BMI (Fig 1). At the 50th percentile, *T. gondii* was associated with an increase of 2.0 units (95% CI [0.22, 3.83]), which represents an increase of 7.3% over the BMI of the 50th-percentile female in poverty (BMI = 27.5). The difference between females <200% poverty and females >200% poverty was never statistically significant. *T. gondii* exposure was not associated with BMI among males <200% poverty, and the difference between males <200% poverty and males >200% poverty was never statistically significant (Fig 2).

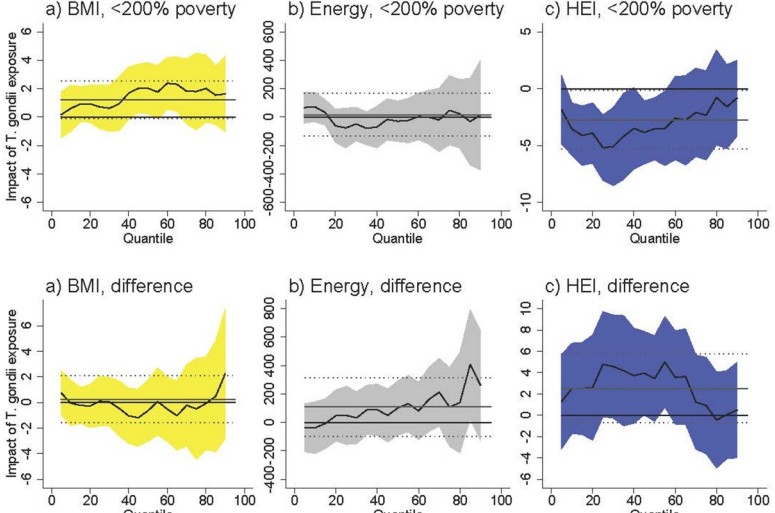

**Fig 1. Association between *T. gondii* exposure and BMI, energy consumed, and HEI for females <200% poverty (a-c) and the difference between females <200% poverty and females >200% poverty ("difference").** Black line is quantile regression coefficient and confidence intervals (95%) are shaded regions. Solid gray line shows the linear regression coefficient and dotted gray lines show the linear regression confidence interval (95%). The dotted line indicates no relationship at that quantile of BMI, energy consumed, and HEI. Straight black line shows value of 0, or no relationship. Controls: income >200% poverty, gender, race/ethnicity (non-Hispanic white, non-Hispanic black, Hispanic, other race), household size, household size squared, age, age squared, whether married, education (less than high school, high school graduate, some college, college graduate), season of interview. Energy and HEI analyses also control for BMI.

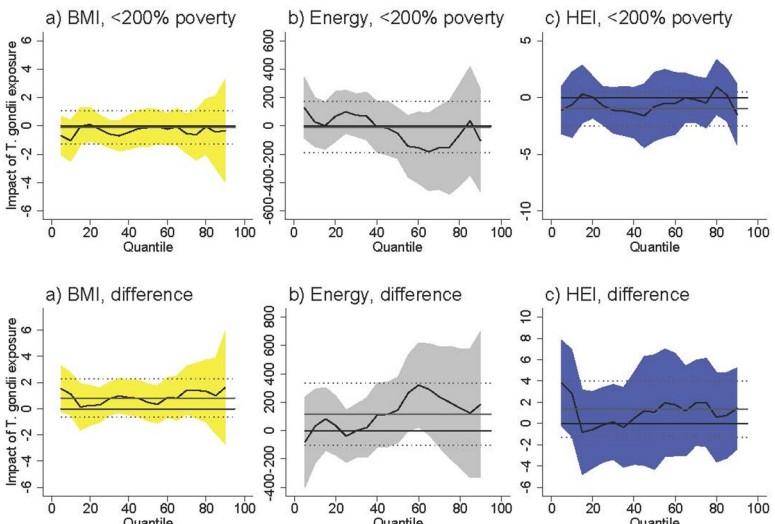

**Fig 2. Association between *T. gondii* exposure and BMI, energy consumed, and HEI for males <200% poverty (a-c) and the difference between males <200% poverty and males >200% poverty ("difference").** Black line is quantile regression coefficient and confidence intervals (95%) are shaded regions. Solid gray line shows the linear regression coefficient and dotted gray lines show the linear regression confidence interval (95%). The dotted line indicates no relationship at that quantile of BMI, energy consumed, and HEI. Straight black line shows value of 0, or no relationship. Controls: income >200% poverty, gender, race/ethnicity (non-Hispanic white, non-Hispanic black, Hispanic, other race), household size, household size squared, age, age squared, whether married, education (less than high school, high school graduate, some college, college graduate), season of interview. Energy and HEI analyses also control for BMI.

*T. gondii* exposure was not associated with energy (kcal) consumption among females <200% of poverty (Fig 1) and among males <200% of poverty (Fig 2). The difference between both females and males <200% of poverty and respondents >200% of poverty is positive and occasionally statistically significant at above-median levels of energy consumption. The coefficients in our models are noisy, but this suggests that *T. gondii* exposure is associated with higher energy consumption among respondents >200% of poverty.

*T. gondii* exposure was associated with lower HEI among females <200% poverty with already-poor diets (i.e. females at the lower end of the HEI distribution) (Fig 1). At the 25th percentile of HEI, *T. gondii* exposure was associated with a lower HEI by 5.05 units [95% CI [-7.87, -2.24]), which represents an 11.7% decrease in HEI score relative to the 25th percentile HEI among females in poverty (43.1). The difference between females <200% of poverty and females >200% of poverty was positive and occasionally statistically significant at lower levels of HEI. Taking into account the negative relationship among females in poverty, this difference suggests there is no relationship between *T. gondii* and HEI among females >200% of poverty. *T. gondii* exposure was not associated with HEI among males <200% of poverty, and the difference between males <200% of poverty and males >200% of poverty is never statistically significant.

## Potential mechanisms

While we found that *T. gondii* exposure is associated with higher BMI and worse diet among females, our data are cross-sectional and do not allow us to make any claim of causality. Therefore, in this section we explore potential mechanisms for our findings among females. We start with potential reasons for a causal relationship and then explore mechanisms that may confound estimation of this causal relationship.

The most direct mechanism explaining our results is that *T. gondii* influences food preferences to facilitate *T. gondii* transmission. Parasites have been observed in multiple contexts changing the host's behavior in order to enhance transmission [40–42], though the extent to which *T. gondii* influences human behavior is controversial [43]. An alternative mechanism is that *T. gondii* may have an indirect effect on food preferences through increasing the host's willingness to undertake risky behavior or by influencing the host's mental health. *T. gondii* exposure has been associated with impulsive or risky behavior in humans [8–11]. This impulsive behavior may translate into food choices that prioritize short-term satisfaction over long-term health. On the other hand, less impulsive behavior may contribute to eating healthier but harder-to-prepare foods, increasing risk of *T. gondii* transmission. In addition to risky behavior, *T. gondii* exposure has been found to be associated with mental health [7,44], which may in turn influence diet-related outcomes. The above mechanisms would explain a direct or indirect causal relationship between *T. gondii* exposure and diet-related behavior.

Beyond a direct or indirect impact of *T. gondii* on diet behavior, our results may be explained by factors that confound estimation of a causal relationship. There are three major sources of confounding factors: food and drink sources, cat and environmental exposure, and coinfections. First, individuals may ingest *T. gondii* through food or drink sources. If individuals with higher BMI or lower HEI are more likely to eat infected foods or prepare food improperly (i.e. not wash produce, not cook meat to appropriate temperatures), they may have greater exposure to *T. gondii*. One important food source of *T. gondii* is infected meat [3]. Higher meat consumption and/or improperly cooking meat could lead to an increased risk of *T. gondii* exposure. In addition, *T. gondii* prevalence differs by type of meat. For example, pork is more likely to contain *T. gondii* than beef [45].

We used multiple methods to test the extent to which meat consumption is a confounding factor. Specifically, we evaluated the association between *T. gondii* exposure and total protein consumption across the protein consumption distribution (Fig 3). We would expect this association to be positive if *T. gondii* seropositive individuals have greater exposure to *T. gondii*

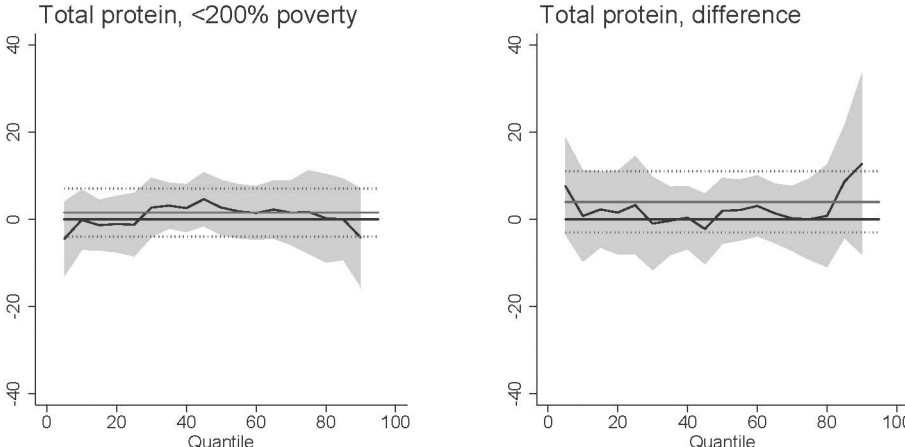

**Fig 3. Association between *T. gondii* exposure and total protein consumption among U.S. females <200% poverty (a-c) and the difference between females <200% poverty and females >200% poverty ("difference").** Black line is quantile regression coefficient and confidence intervals (95%) are shaded regions. Solid gray line shows the linear regression coefficient and dotted gray lines show the linear regression confidence interval (95%). The dotted line indicates no relationship at that quantile of BMI, energy consumed, and HEI. Straight black line shows value of 0, or no relationship. Controls: income >200% poverty, gender, race/ethnicity (non-Hispanic white, non-Hispanic black, Hispanic, other race), household size, household size squared, age, age squared, whether married, education (less than high school, high school graduate, some college, college graduate), season of interview.

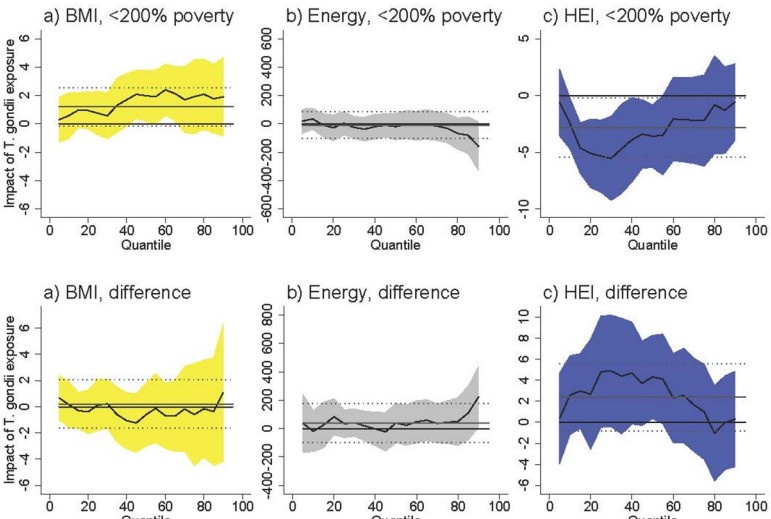

**Fig 4. Association between *T. gondii* exposure and diet-related outcomes among U.S. females <200% poverty (a-c) and the difference between females <200% poverty and males >200% poverty ("difference"), controlling for total protein consumption.** Black line is quantile regression coefficient and confidence intervals (95%) are shaded regions. Solid gray line shows the linear regression coefficient and dotted gray lines show the linear regression confidence interval (95%). The dotted line indicates no relationship at that quantile of BMI, energy consumed, and HEI. Straight black line shows value of 0, or no relationship. Controls: total protein consumption, income >200% poverty, gender, race/ethnicity (non-Hispanic white, non-Hispanic black, Hispanic, other race), household size, household size squared, age, age squared, whether married, education (less than high school, high school graduate, some college, college graduate), season of interview. Energy and HEI analyses also control for BMI.

through higher levels of meat consumption. Instead, this association was statistically insignificant at all ranges of the protein consumption distribution. Furthermore, our main results were robust to controlling for protein consumption ([Fig 4]). While total meat consumption does not substantially confound our results, the composition of meat consumed by *T. gondii* seropositive individuals may be different than seronegative individuals. In particular, *T. gondii* seropositive individuals may eat more pork and thus be exposed to *T. gondii* ([Table 1]). *T. gondii* exposure had no statistically significant association with the mean percent of energy coming from pork (column 1), whether the individual ate any pork (column 2), or whether pork consumption was in higher ends of the pork consumption distribution (columns 3–5). Our main

**Table 1. Association between *T. gondii* exposure and pork consumption.**

|  | (1) | (2) | (3) | (4) | (5) |
|---|---|---|---|---|---|
|  | **% pork** | **Any pork** | **Pork 1–11%** | **Pork 12–20%** | **Pork >20%** |
| T. gondii exposure, <200% poverty | 0.0002 | -0.0111 | -0.0139 | 0.0053 | -0.0025 |
|  | (0.0023) | (0.0121) | (0.0072) | (0.0078) | (0.0038) |
| Difference | 0.0032 | 0.0210 | 0.0095 | 0.0046 | 0.0070 |
|  | (0.0047) | (0.0224) | (0.0103) | (0.0145) | (0.0106) |

Coefficients are from linear regressions of the percent of total energy consumption (kcal) from pork products (column 1), an indicator for whether any pork was consumed (column 2), an indicator for whether pork was 1–11% of energy consumption (column 3), an indicator for whether pork was 12–20% of energy consumption (column 4), or an indicator for whether pork was >20% of energy consumption (column 5) on *T. gondii* exposure and controls. Cutoffs of 11% and 20% represent the 50[th] and 75[th] percentiles of the non-zero pork consumption distribution. Standard errors in parentheses. Bold coefficients are significant at the 95% confidence level. Controls: income >200% poverty, gender, race/ethnicity (non-Hispanic white, non-Hispanic black, Hispanic, other race), household size, household size squared, age, age squared, whether married, education (less than high school, high school graduate, some college, college graduate), season of interview, BMI.

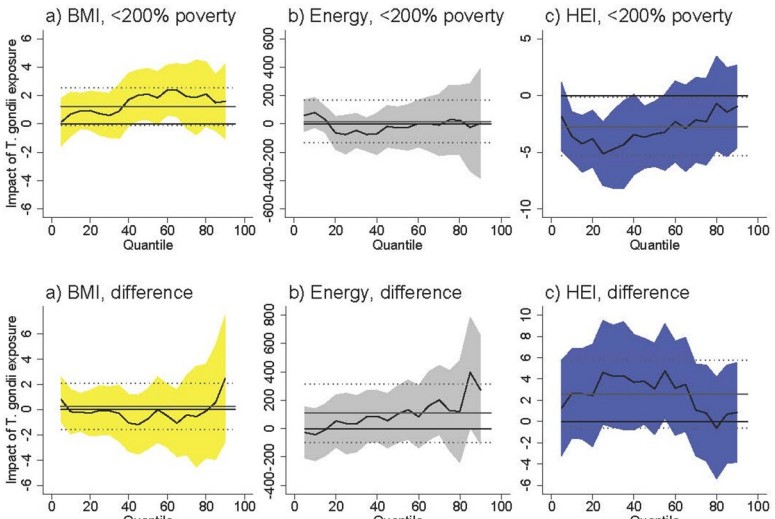

**Fig 5. Association between *T. gondii* exposure and diet-related outcomes among U.S. females <200% poverty (a-c) and the difference between females <200% poverty and males >200% poverty ("difference"), controlling for % of total energy consumption from pork.** Black line is quantile regression coefficient and confidence intervals (95%) are shaded regions. Solid gray line shows the linear regression coefficient and dotted gray lines show the linear regression confidence interval (95%). The dotted line indicates no relationship at that quantile of BMI, energy consumed, and HEI. Straight black line shows value of 0, or no relationship. Controls: % of total energy consumption from pork, income >200% poverty, gender, race/ethnicity (non-Hispanic white, non-Hispanic black, Hispanic, other race), household size, household size squared, age, age squared, whether married, education (less than high school, high school graduate, some college, college graduate), season of interview. Energy and HEI analyses also control for BMI.

results were also robust to controlling for the percent of total energy consumption coming from pork (Fig 5). Differential total protein consumption and the composition of current meat consumption thus do not explain our main results. Notably, consideration of protein consumption does not account for potential differences in the individual's *history* of protein consumption nor the proper *preparation* of meat. If food preparation drives our findings, then we would expect higher *T. gondii* infection among females since U.S. females are more likely than males to cook [46]. However, more males than females have evidence of *T. gondii* infection in our sample (13.1% of males and 12.1% of females).

In addition to meat, *T. gondii* has also been found on produce and in drinking water [47–48]. Diets that include significant amounts of produce or contaminated water may therefore have higher *T. gondii* exposure. Since by construction more produce consumption *increases* the HEI score, we would expect to find *T. gondii* infection associated with an increase in HEI and possibly lower BMI. However, we found the opposite relationship, suggesting that the amount of produce consumed may not be an important mechanism. The possibility remains that the produce consumed by individuals with lower HEI scores is also more likely to be contaminated. Different exposure to *T. gondii* can also come from different drinking water sources, with water from wells is substantially more likely to be contaminated by *T. gondii* than water from municipal sources [47]. However, controlling whether the respondent uses well water (vs. a municipal source) did not change our main results (Fig 6), suggesting that differences in drinking water sources are unlikely to explain our findings.

Second, individuals can be exposed to *T. gondii* through sources besides food and water. *T. gondii* reproduces in cats and spreads into the environment through cat feces. Locations where cats choose to defecate, such as sandboxes and gardens, are at particular risk for having high concentrations of *T. gondii* and can lead to infection among individuals who live near these

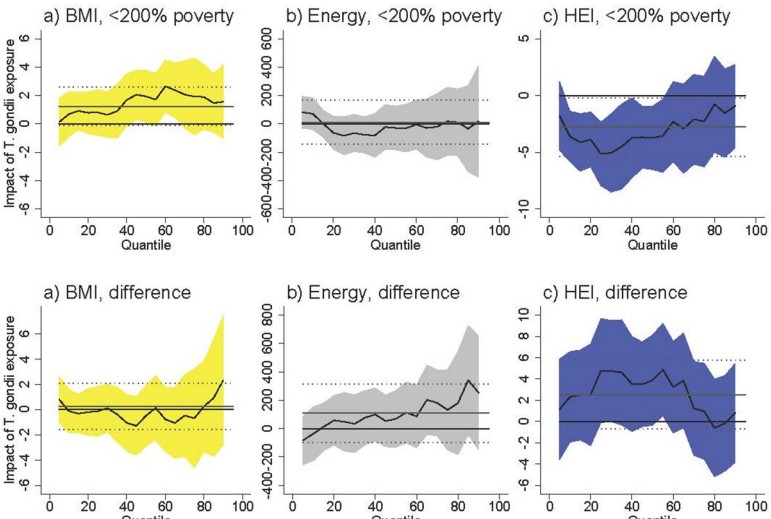

**Fig 6. Association between *T. gondii* exposure and diet-related outcomes among U.S. females <200% poverty (a-c) and the difference between females <200% poverty and males >200% poverty ("difference"), controlling for water source.** Black line is quantile regression coefficient and confidence intervals (95%) are shaded regions. Solid gray line shows the linear regression coefficient and dotted gray lines show the linear regression confidence interval (95%). The dotted line indicates no relationship at that quantile of BMI, energy consumed, and HEI. Straight black line shows value of 0, or no relationship. Controls: tap water from well water, income >200% poverty, gender, race/ethnicity (non-Hispanic white, non-Hispanic black, Hispanic, other race), household size, household size squared, age, age squared, whether married, education (less than high school, high school graduate, some college, college graduate), season of interview. Energy and HEI analyses also control for BMI.

locations [48]. Cat ownership information was not available in the NHANES waves that measure *T. gondii* exposure, so we were unable to test whether previous exposure to cats moderates the relationship between *T. gondii* exposure and diet. However, cat ownership information was recorded in the 2005–2006 NHANES wave. Fig 7 displays the association between current cat ownership and our diet-related outcomes using the 2005–2006 NHANES wave. These associations were smaller than the main effects of *T. gondii* exposure and were statistically insignificant.

Since cats often live outside and defecate in areas where non-owners frequent, many non-owners can have significant exposure to cats as well. Our results may be explained by differential environmental exposure if individuals with worse diets or higher BMI choose to frequent areas with greater exposure to outdoor cats. While it is unlikely that the presence of outdoor cats directly influences an individual's choice of area, it is unclear what neighborhood characteristics are related to the presence of outdoor cats. Furthermore, if such residential sorting is a factor, it is unclear why we would observe a relationship for females and not for males. We were unable to examine differential environmental exposure using the NHANES.

The final mechanism explaining our results is the possibility of coinfection with other diseases. Coinfection may cause a researcher to attribute effects to one infection that are properly attributed to another infection. In addition, infections may interact and cause behavioral or biological changes different from those of either infection individually. In particular, the parasites *Toxocara canis* and *Toxocara cati* cause toxocariasis and transmission of *T. canis* and *T. cati* is similar to that of *T. gondii* [49–50]. Notably, toxocariasis is associated with worse cognitive function among children [51]. Fig 8 displays our main results controlling for *Toxocara* serostatus. Including this control does not change our main results, indicating that our main

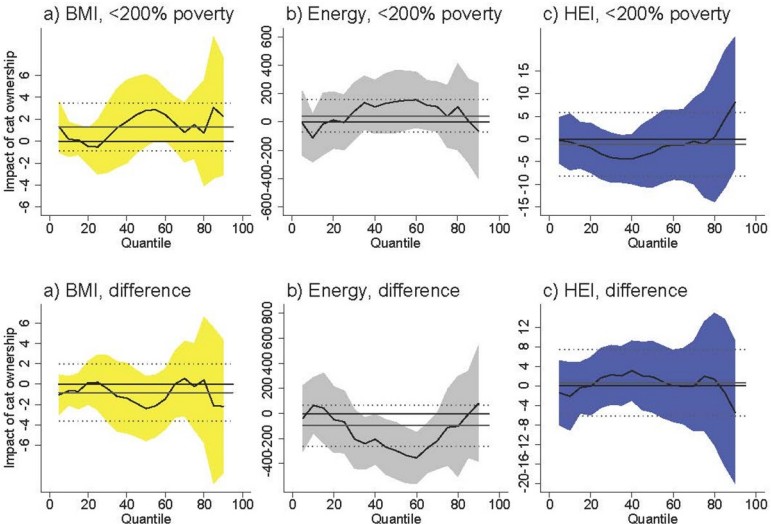

**Fig 7. Association between cat ownership and diet-related outcomes among U.S. females <200% poverty (a-c) and the difference between females <200% poverty and males >200% poverty ("difference").** Black line is quantile regression coefficient and confidence intervals (95%) are shaded regions. Solid gray line shows the linear regression coefficient and dotted gray lines show the linear regression confidence interval (95%). The dotted line indicates no relationship at that quantile of BMI, energy consumed, and HEI. Straight black line shows value of 0, or no relationship. Controls: income >200% poverty, gender, race/ethnicity (non-Hispanic white, non-Hispanic black, Hispanic, other race), household size, household size squared, age, age squared, whether married, education (less than high school, high school graduate, some college, college graduate), season of interview. Energy and HEI analyses also control for BMI.

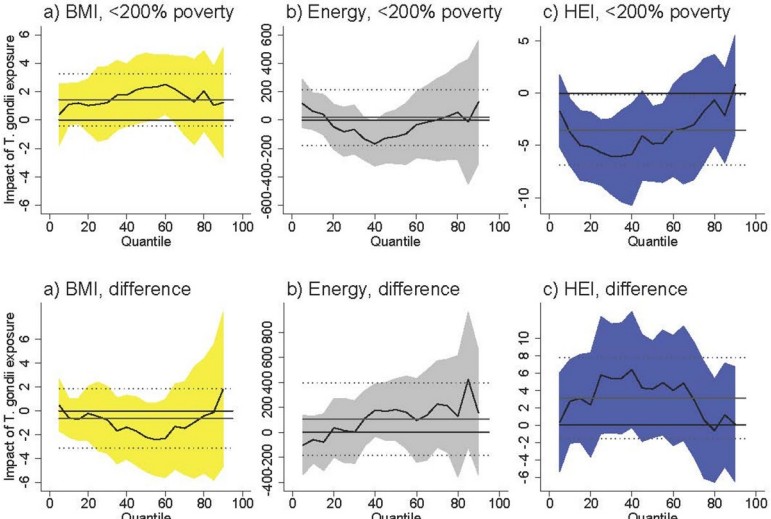

**Fig 8. Association between *T. gondii* exposure and diet-related outcomes among U.S. females <200% poverty (a-c) and the difference between females <200% poverty and males >200% poverty ("difference"), controlling for *Toxocara* serostatus.** Black line is quantile regression coefficient and confidence intervals (95%) are shaded regions. Solid gray line shows the linear regression coefficient and dotted gray lines show the linear regression confidence interval (95%). The dotted line indicates no relationship at that quantile of BMI, energy consumed, and HEI. Straight black line shows value of 0, or no relationship. Controls: *Toxocara* seropositive, income >200% poverty, gender, race/ethnicity (non-Hispanic white, non-Hispanic black, Hispanic, other race), household size, household size squared, age, age squared, whether married, education (less than high school, high school graduate, some college, college graduate), season of interview. Energy and HEI analyses also control for BMI.

results are not driven by coinfection with *Toxocara*. While we cannot rule out other coinfections, we note that *Toxocara* is a common infectious disease in the US [2,49].

This section has discussed potential mechanisms for observing a relationship between *T. gondii* exposure and diet-related outcomes. Beyond these mechanisms, the potential exists for our results to be driven by sample selection. NHANES examination, dietary, and *T. gondii* data are recorded for most, but not all, of the NHANES respondents. As noted above, only samples with surplus serum were tested for *T. gondii* antibodies. BMI data are available for respondents with valid height and weight data, and diet information (total energy consumption and HEI) is available for NHANES respondents who completed the food intake module. If diet-related outcomes among respondents with *T. gondii* data were systematically different from outcomes among respondents without *T. gondii* data, our results may be an artifact of this selection. Out of the total 13,082 respondents in our survey waves, 1,916 have missing *T. gondii* exposure information, 601 respondents have missing BMI data, 1,224 have missing energy consumption information, and 1,224 have missing HEI data. All respondents with missing energy consumption have missing HEI scores and vice versa. Table 2 tests whether *T. gondii* or our outcomes are systematically missing. We found that none of our main outcomes (BMI, energy consumption, HEI) predict whether a respondent has missing *T. gondii* data. Likewise, *T. gondii* exposure does not predict having missing BMI or diet data. These results suggest that patterns of missing values are not driving our results.

Our main results found that *T. gondii* exposure was related with increased BMI at higher ranges of the BMI distribution. While BMI is commonly used in public health studies, BMI may mis-identify as obese individuals with greater muscle mass [52]. We investigated the extent to which our results are influenced by this potential bias in three ways. First, we excluded individuals who report participating in vigorous recreational activities at least three days per week. Excluding these highly-active individuals, our results remain robust (Fig 9). Second, we performed our analysis on the waist-to-height ratio (WHtR) instead of BMI. The WHtR has been found to predict health risk and mortality better than BMI [53–54]. Similar to our main results, *T. gondii* is associated with increased WHtR among females with higher WHtR (Fig 10). Finally, the NHANES recorded the percent body fat for 3,668 respondents out of our total sample. We replicated our analysis with this subsample using percent body fat instead of BMI (Fig 11). We do not emphasize these results because of the small sample size and because fewer respondents with higher levels of BMI have valid percent body fat measures [55]. However, *T. gondii* exposure is related with increased percent body fat among females with median percent body fat.

## Conclusions

Our results uncover important relationships between *T. gondii* exposure and diet-related outcomes for females. Overall, the relationship with BMI and diet quality was strongest for females with high BMIs and worse diets. Other studies have observed similar patterns where obesity determinants have stronger relationships with BMI at increasingly higher levels of BMI [35–36]. On the other hand, we found insignificant relationships between *T. gondii* exposure and diet/health outcomes among males. Our study suggests that exposure to a largely apathogenic agent could have broader consequences for individual and population health.

We also examined potential mechanisms underlying our findings. Some of these mechanisms represent factors that confound a causal interpretation of our findings. While we were able to provide evidence that our findings were not fully explained by some mechanisms, other mechanisms remained unexplored. This represents one limitation of our study. Data availability does not allow us to examine many sources of exposure to *T. gondii*. Important potential

**Table 2. Determinants of missing *T. gondii*, BMI, and HEI data.**

| Variable | Outcome *T. gondii* | BMI | HEI |
|---|---|---|---|
| BMI | -0.0005 | | |
| | (0.0004) | | |
| Energy (kcal) | -0.000003 | | |
| | (0.000003) | | |
| HEI | -0.0003 | | |
| | (0.0002) | | |
| T. gondii exp. | | -0.0025 | -0.0081 |
| | | (0.0034) | (0.0066) |
| Black | **0.0864** | -0.0027 | **0.0203** |
| | (0.0070) | (0.0027) | (0.0053) |
| Hispanic | 0.0123 | -0.0054 | 0.0058 |
| | (0.0093) | (0.0035) | (0.0069) |
| Other race/ethnicity | 0.0185 | 0.0019 | 0.0087 |
| | (0.0135) | (0.00512) | (0.0101) |
| Household size | 0.0001 | 0.0004 | 0.0047 |
| | (0.0082) | (0.0031) | (0.0061) |
| HH size squared | 0.0002 | 0.00010 | -0.00003 |
| | (0.0011) | (0.0004) | (0.0008) |
| Age | -0.0010 | -0.0011 | -0.0012 |
| | (0.0009) | (0.0003) | (0.0007) |
| Age squared | 0.000007 | **0.00002** | 0.00002 |
| | (0.000009) | (0.000003) | (0.000007) |
| >200% of poverty ratio | 0.0145 | -0.0042 | -0.0002 |
| | (0.0062) | (0.0024) | (0.0046) |
| Married | -0.0151 | -0.0051 | **-0.0188** |
| | (0.0069) | (0.0026) | (0.0051) |
| Interview between Nov. and Apr. | 0.0031 | -0.0016 | -0.0004 |
| | (0.0058) | (0.0022) | (0.0043) |
| Education: High school | -0.0111 | -0.0051 | -0.0045 |
| | (0.0086) | (0.0033) | (0.0064) |
| Education: Some college | -0.0213 | -0.0036 | -0.0052 |
| | (0.0081) | (0.0031) | (0.0060) |
| Education: college or higher | -0.0182 | -0.0033 | 0.0016 |
| | (0.0095) | (0.0036) | (0.0070) |
| Constant | **0.1510** | **0.0343** | 0.0532 |
| | (0.0293) | (0.0097) | (0.0189) |

Table displays linear probability model coefficients for determinants of an NHANES respondent having missing *T. gondii* data, BMI data, or HEI data. Standard errors in parentheses. Bold coefficients are significant at the 95% confidence level. Controls: income >200% poverty, gender, race/ethnicity (non-Hispanic white, non-Hispanic black, Hispanic, other race), household size, household size squared, age, age squared, whether married, education (less than high school, high school graduate, some college, college graduate), season of interview.

sources that we cannot observe in our data include contamination of meat or produce and environmental exposure. Since we cannot rule out other mechanisms, we are not able to establish a causal relationship between *T. gondii* exposure and diet-related measures. This broader limitation is common in cross-sectional observational studies and provides directions for

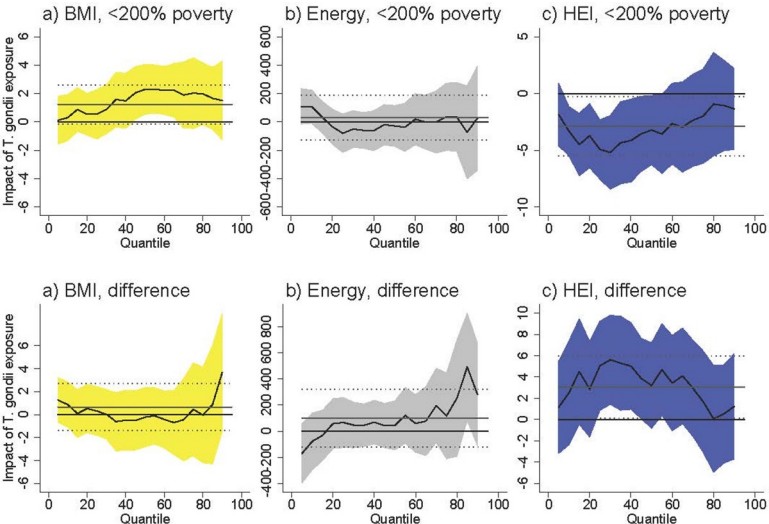

**Fig 9. Association between *T. gondii* exposure and diet-related outcomes among U.S. females <200% poverty (a-c) and the difference between females <200% poverty and males >200% poverty ("difference"), excluding highly-active individuals.** Black line is quantile regression coefficient and confidence intervals (95%) are shaded regions. Solid gray line shows the linear regression coefficient and dotted gray lines show the linear regression confidence interval (95%). The dotted line indicates no relationship at that quantile of BMI, energy consumed, and HEI. Straight black line shows value of 0, or no relationship. Controls: income >200% poverty, gender, race/ethnicity (non-Hispanic white, non-Hispanic black, Hispanic, other race), household size, household size squared, age, age squared, whether married, education (less than high school, high school graduate, some college, college graduate), season of interview. Energy and HEI analyses also control for BMI.

further research. A full investigation of these mechanisms would proceed on two fronts. First, a more detailed understanding of environmental exposure to *T. gondii* is needed. This analysis would entail examining the extent to which *T. gondii* exposure is influenced by food

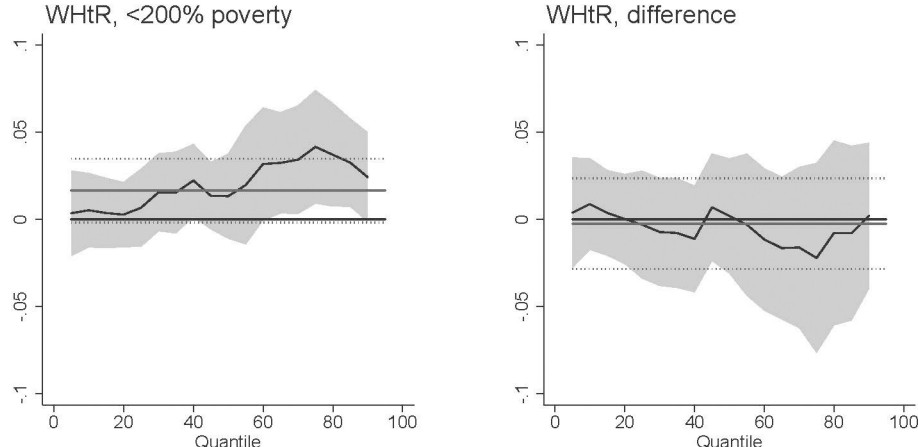

**Fig 10. Association between *T. gondii* exposure and waist-to-height ratio (WHtR) among U.S. females <200% poverty (a-c) and the difference between females <200% poverty and males >200% poverty ("difference").** Black line is quantile regression coefficient and confidence intervals (95%) are shaded regions. Solid gray line shows the linear regression coefficient and dotted gray lines show the linear regression confidence interval (95%). The dotted line indicates no relationship at that quantile of BMI, energy consumed, and HEI. Straight black line shows value of 0, or no relationship. Controls: income >200% poverty, gender, race/ethnicity (non-Hispanic white, non-Hispanic black, Hispanic, other race), household size, household size squared, age, age squared, whether married, education (less than high school, high school graduate, some college, college graduate), season of interview.

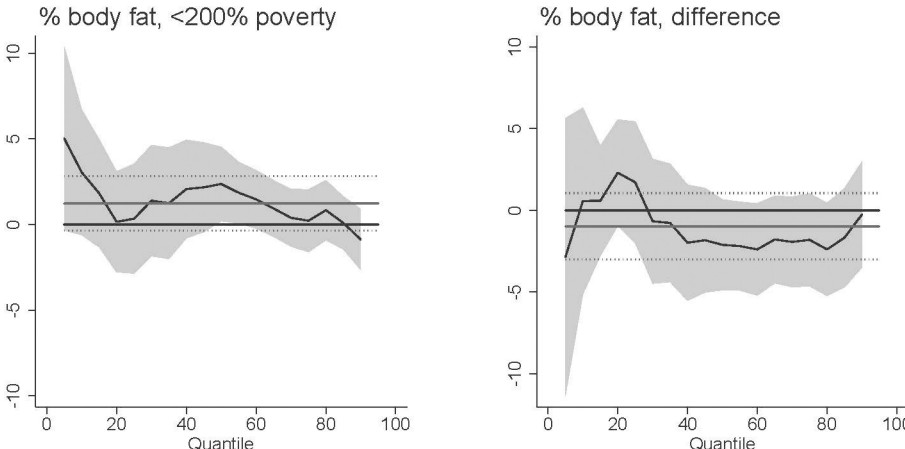

**Fig 11. Association between *T. gondii* exposure and percent body fat among U.S. females <200% poverty (a-c) and the difference between females <200% poverty and males >200% poverty ("difference").** Black line is quantile regression coefficient and confidence intervals (95%) are shaded regions. Solid gray line shows the linear regression coefficient and dotted gray lines show the linear regression confidence interval (95%). The dotted line indicates no relationship at that quantile of BMI, energy consumed, and HEI. Straight black line shows value of 0, or no relationship. Controls: income >200% poverty, gender, race/ethnicity (non-Hispanic white, non-Hispanic black, Hispanic, other race), household size, household size squared, age, age squared, whether married, education (less than high school, high school graduate, some college, college graduate), season of interview.

preparation practices for both meat and produce. In addition, this analysis would entail an investigation of exposure to outdoor cats and how the prevalence of *T. gondii* in public spaces differs between neighborhoods. Second, research is needed that elucidates the biological mechanisms for the relationship between *T. gondii* exposure and diet-related outcomes. Studies using non-human subjects such as mice may investigate the extent to which *T. gondii* exposure either directly or indirectly leads to changes in eating preferences.

Characterizing how *T. gondii* exposure influences diet is necessary for formulating a comprehensive public health policy. Unhealthy diets and concurrent obesity increase the risk of mortality and morbidity [56–59]. Recent attention has focused on potential biological bases for diets and obesity [60–62], but little is known about how these biological factors interact with other factors to influence diet and health [63]. To the extent that the differences we find uncovered causal relationships, *T. gondii* exposure could be exacerbating the current obesity crisis. *T. gondii* exposure may furthermore work against public health policies that seek to incentivize healthy food consumption. These possibilities suggest the need for making a toxoplasmosis vaccine widely available [64]. Our findings highlight that women with poor diets may benefit most from access to a toxoplasmosis vaccine. The potential effect of *T. gondii* exposure in exacerbating public health challenges–and working against current public health policy–underscores the importance of understanding the mechanisms underlying the relationships found in this paper.

## Author Contributions

**Conceptualization:** Joel Cuffey, Nicholas M. Fountain-Jones.

**Data curation:** Joel Cuffey, Shuoli Zhao.

**Formal analysis:** Joel Cuffey.

**Methodology:** Christopher A. Lepczyk, Shuoli Zhao, Nicholas M. Fountain-Jones.

**Supervision:** Christopher A. Lepczyk.

**Writing – original draft:** Joel Cuffey, Christopher A. Lepczyk, Shuoli Zhao, Nicholas M. Fountain-Jones.

**Writing – review & editing:** Joel Cuffey, Christopher A. Lepczyk, Shuoli Zhao, Nicholas M. Fountain-Jones.

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
