## [Decision Letter · Decision Letter 0]

9 Jul 2021

Dear Dr. Cuffey,

Thank you very much for submitting your manuscript "Toxoplasma gondii exposure and risk for poor diets and higher BMI" for consideration at PLOS Neglected Tropical Diseases. As with all papers reviewed by the journal, your manuscript was reviewed by members of the editorial board and by several independent reviewers. In light of the reviews (below this email), we would like to invite the resubmission of a significantly-revised version that takes into account the reviewers' comments. 

We would like you to please address carefully the comments made by the three reviewers, and particularly discuss the points that were made regarding data limitations, selection and measurement biases, and overall methodology - and what this implies for the validity and transferrability of the results.

The reviewers have raised some concerns about some statements implying causation when the data & methodology does not allow for causal inference. Please revise the manuscript carefully to remove any ambiguity in that regard that may lead to misleading interpretations of the implications of this research. Possible alternative explanations for the relationship observed should be acknowledged and discussed accordingly in the discussion.

We cannot make any decision about publication until we have seen the revised manuscript and your response to the reviewers' comments. Your revised manuscript is also likely to be sent to reviewers for further evaluation.

Sincerely,

Laure Saulais

Guest Editor

Steven Singer

Deputy Editor

I would like to ask the authors to please address carefully the comments made by the three reviewers, and particularly discuss the points that were made regarding data limitations, selection and measurement biases, and overall methodology - and what this implies for the validity and transferrability of the results.

The reviewers have raised some concerns about some statements implying causation when the data & methodology does not allow for causal inference. Please revise the manuscript carefully to remove any ambiguity in that regard that may lead to misleading interpretations of the implications of this research. Possible alternative explanations for the relationship observed should be acknowledged and discussed accordingly in the discussion.

Reviewer's Responses to Questions

**Key Review Criteria Required for Acceptance?**

**Methods**

-Are the objectives of the study clearly articulated with a clear testable hypothesis stated?

-Is the study design appropriate to address the stated objectives?

-Is the population clearly described and appropriate for the hypothesis being tested?

-Is the sample size sufficient to ensure adequate power to address the hypothesis being tested?

-Were correct statistical analysis used to support conclusions?

-Are there concerns about ethical or regulatory requirements being met?

Reviewer #1: The manuscript attempts to study the relationships between diet and Toxoplasma gondii infections, and uses a dataset from the National Health and Nutrition Examination Survey (NHANES) to conduct their analysis. This dataset has limitations, such as:

- biases regarding self-reported eating habits (McKenzie et al., The American Journal of Clinical Nutrition, 2021)

- inability to take into account food safety and hygiene, which are relevant to Toxoplasmosis transmission (Hussain et al., Pathogens, 2017)

- lack of discrimination in the kind of food being consume (for instance, different meats might have different parasite loads (Belluco et al., Plos One, 2016))

- cat ownership data only takes into account current ownership and/or a cat that lived in the same household for the past 12 months. The serological test used in the study detects IgG, which usually last for many years after infection. Additionally, correlate cat ownership and toxoplasma positivity would further strenght the data presented in tables S7 and S8

Reviewer #2: Yes to all those questions.

Reviewer #3: 1- BMI and WHR have high false-positive rates in reference to BFP, which cannot accurately reflect the mass of adipose tissue and leads to obesity misclassification. Pregnant women were excluded from the study, but not athletes and people with high muscle mass who had a false-positive high BMI.

Please explain about this factor and how to eliminate its effect on the results.

2- The level of physical activity is one of the factors affecting the BMI. Please explain about matching the BMi results with it.

**Results**

-Does the analysis presented match the analysis plan?

-Are the results clearly and completely presented?

-Are the figures (Tables, Images) of sufficient quality for clarity?

Reviewer #1: (No Response)

Reviewer #2: Yes to all those questions.

Reviewer #3: 1- Please correct numbers in tables by the following guidelines are usually applicable.

*One decimal place 

 -Means

 -Standard deviations

 -Descriptive statistics based on discrete data

*Two decimal places

 -Correlation coefficients

 -Proportions

 -Inferential test statistics such as t values, F values, and chi-squares

*Use two or three decimal places and report exact values for all p values greater than .001. For p values smaller than .001, report them as p < .001.

**Conclusions**

-Are the conclusions supported by the data presented?

-Are the limitations of analysis clearly described?

-Do the authors discuss how these data can be helpful to advance our understanding of the topic under study?

-Is public health relevance addressed?

Reviewer #1: Although the extent in which T. gondii affects rodent behavior is still a topic of debate (Worth et al., Advances in Parasitology, 2014), the large body of literature on the subject suggest that these changes increase the likelihood of feline predation (for instance, Vyas et al., PNAS, 2007). These behavioral changes makes sense in light of evolution, as the parasite uses rodents and felines as intermediate and definitive hosts in its life cycle, and contact between the parasite and humans occured very recently (Gazzinelli et al., Cell Host and Microbe, 2014) (Shwab et al., PNAS, 2018). The links between T. gondii and changes in human behavior are controversial (Johnson et al., mBio, 2020) (Sugden et al., Plos One, 2016) and claims linking the parasite to behavioral changes in humans need to be backed by a substantial body of data. The data presented in the article correlating T. gondii infection and bmi/hei cannot be used to imply causation, as a more plausible explanation would be that higher food consuption and/or poor quality of food increases the likelihood of T. gondii infection.

Reviewer #2: Yes to all those questions.

Reviewer #3: Researchers have noted in various articles on toxoplasmosis exposure as causing high-risk and impulsive behaviors. It should be noted that perhaps reversing high-risk, impulsive behaviors may increase the risk of developing or being exposed to toxoplasmosis in these individuals.

please discuss about this issue

**Editorial and Data Presentation Modifications?**

Reviewer #1: (No Response)

Reviewer #2: MINOR COMMENTS:

1. Title: I suggest revising to “Cross-sectional association of Toxoplasma gondii exposure with BMI and diet in US adults.” (Some readers may misinterpret the current title as indicating a longitudinal study of the emergence of higher BMI or poor diet over time after T. gondii exposure.)

2. Abstract: Consider reporting 95% CIs for the statistics included in the abstract. Also, consider including more numerical results in the abstract. Finally, consider mentioning the rationale for analyzing women and men separately, and for analyzing low-income people separately, in the abstract, given that results are presented separately by those characteristics.

3. Introduction, paragraph 2: Consider specifying what country was the setting for reference #22 rather than referring to “not … all populations.”

4. Methods, statistical analysis: Why did the authors adjust for season of interview? Consider providing a rationale in the manuscript.

5. Tables/figures: Consider listing the control variables in a footnote below each table or figure so that the reader does not need to refer back to the Methods section to see what was adjusted for.

Reviewer #3: N/A

**Summary and General Comments**

Reviewer #1: The study cannot imply causation from correlation, and more robust data would be required to imply that a parasite causes changes in human behavior, specially considering that humans are accidental hosts.

Reviewer #2: The authors studied the associations of Toxoplasma gondii exposure with BMI, total energy intake, and Healthy Eating Index (HEI) in a cross-sectional study of 9,853 American-born women and men in NHANES. They found that T. gondii exposure was associated with higher BMI in women, not in men, and with lower HEI in low-income women, not in all women or in men. Investigation of some potential biological mechanisms including protein consumption, water source, cat ownership, and Toxocara coinfection suggested that such mechanisms do not explain the observed associations.

In my opinion, the authors based their study on a strong rationale, they expressed their research questions clearly, their approach was methodologically strong in most respects, their reporting of results was thorough and well organized, and their conclusions were justified by the results. The manuscript was well written.

I have a few comments and questions for the authors to consider: 

MAJOR COMMENTS:

1. Methods: Regarding the potential for selection bias, I have four questions: (1) How many people from the total 2009-2014 sample were excluded from analysis due to missing T. gondii exposure? (2) How many were excluded due to missing BMI or diet data? (3) To what degree did exclusion for missing T. gondii measure differ across BMI, total energy, or HEI values? (4) To what degree did exclusion for missing BMI or dietary data differ by T. gondii exposure? Reporting this information in the manuscript would help some readers better assess the potential for selection bias affecting the results, and could also allow the authors to comment more about that possibility in the discussion.

2. Methods/Results: In the investigation of heterogeneity by low-income status, consider reporting results for those above 200% of the poverty line in addition to those below 200% of the poverty line, so that readers will be able to see both sides. Also, consider investigating the T. gondii by low-income interaction in a single model with an interaction term and calculating the excess association due to interaction. Such an analysis would better support the authors’ interest in the low-income population by determining the degree to which the joint effect of T. gondii plus low-income is stronger than would be expected if the two exposures didn’t interact.

3. Methods, statistical analysis: Why did the authors analyze females and males separately, with no models combining females and males? Consider providing a rationale in the manuscript. Was this decision made a priori or influenced by the data analysis?

4. Tables: Consider reporting 95% CI for each estimate instead of standard error. Also consider showing the results for the group that is above 200% of the poverty line.

5. Figures: Consider using the same Y-axis scales in the figures for women and for men, in order to make the visual comparison easier. Also consider showing the graphs for the group that is above 200% of the poverty line.

6. Results, potential mechanisms: I found this section easy to understand and logically organized, however the organizational scheme was unusual in the context of the overall IMRaD (Intro-Methods-Results-and-Discussion) structure. A lot of sentences in this section of Results sounded more like Intro, Methods, and Discussion. Consider moving sentences into the appropriate sections of the article. However, you may want to consult with the journal’s editorial team on this.

7. Supplemental tables and text corresponding with Results, potential mechanisms: First, I recommend incorporating all supplemental text into the main body of the manuscript in appropriate places (some may fit better in Intro, Methods, or Discussion, than in Results, similar to above comment). Second, I recommend replacing all Supplemental Tables with figures similar to the figures shown in the main manuscript. I think all the points the authors want to make in this section about potential mechanisms will be made more clear with visual figures instead of lots of tables. Finally, if the authors accept my suggestion to replace the supplemental tables with figures, consider incorporating some (or all) of the new figures into the main manuscript instead of a supplement. The exploration of potential mechanisms was one of the most interesting parts of the article, and in my view deserves greater emphasis and visibility. To make room for more figures in the main manuscript, consider moving Tables 1 and 2 into a supplement. The figures really showcase the findings better than the tables, in my opinion.

8. Discussion: The cross-sectional study design is not the strongest for this type of research question, where the hypothesis is that T. gondii exposure LEADS TO increased BMI or deterioration in diet quality, but the data structure do not permit knowing which came first, the T. gondii exposure versus the BMI and diet quality levels. Although the authors mention the limitations of cross-sectional studies, I suggest stating more clearly and strongly the difficulty in drawing causal inference from this particular study on this particular question. To what extent is reverse-causality a possible explanation for the results?

MINOR COMMENTS -- See above in the section "Editorial and Data Presentation Modifications?"

I prefer to sign my review – Evan Thacker, PhD, Brigham Young University

Reviewer #3: The goals and methodology of this study is noteworthy, although the authors need to address some minor problems in the methodology and reporting data as well as discussion section.

PLOS authors have the option to publish the peer review history of their article (what does this mean?). If published, this will include your full peer review and any attached files.

Reviewer #1: No

Reviewer #2: Yes: Evan L. Thacker

Reviewer #3: Yes: Reza HabibiSaravi, MD, PhD

Mazandaran University of Medical Sciences
---

## [Decision Letter · Decision Letter 1]

20 Sep 2021

Dear Dr. Cuffey,

We are pleased to inform you that your manuscript 'Cross-sectional association of Toxoplasma gondii exposure with BMI and diet in US adults' has been provisionally accepted for publication in PLOS Neglected Tropical Diseases.

Best regards,

Laure Saulais

Guest Editor

Steven Singer

Deputy Editor

Reviewer's Responses to Questions

**Key Review Criteria Required for Acceptance?**

**Methods**

-Are the objectives of the study clearly articulated with a clear testable hypothesis stated?

-Is the study design appropriate to address the stated objectives?

-Is the population clearly described and appropriate for the hypothesis being tested?

-Is the sample size sufficient to ensure adequate power to address the hypothesis being tested?

-Were correct statistical analysis used to support conclusions?

-Are there concerns about ethical or regulatory requirements being met?

Reviewer #1: (No Response)

Reviewer #3: Yes

**Results**

-Does the analysis presented match the analysis plan?

-Are the results clearly and completely presented?

-Are the figures (Tables, Images) of sufficient quality for clarity?

Reviewer #1: (No Response)

Reviewer #3: Yes

**Conclusions**

-Are the conclusions supported by the data presented?

-Are the limitations of analysis clearly described?

-Do the authors discuss how these data can be helpful to advance our understanding of the topic under study?

-Is public health relevance addressed?

Reviewer #1: (No Response)

Reviewer #3: Yes

**Editorial and Data Presentation Modifications?**

Reviewer #1: (No Response)

Reviewer #3: Accept

**Summary and General Comments**

Reviewer #1: I believe the modifications greatly improved the manuscript. The revised manuscript is very clear and well written.

Reviewer #3: This study is now as good as for publication.

PLOS authors have the option to publish the peer review history of their article (what does this mean?). If published, this will include your full peer review and any attached files.

Reviewer #1: No

Reviewer #3: **Yes: **Reza HabibiSaravi, MD, PhD

Mazandaran University of Medical Sciences

---

## [Editor Report · Acceptance letter]

27 Sep 2021

Dear Dr. Cuffey,

We are delighted to inform you that your manuscript, "Cross-sectional association of Toxoplasma gondii exposure with BMI and diet in US adults," has been formally accepted for publication in PLOS Neglected Tropical Diseases.

Best regards,

Shaden Kamhawi

co-Editor-in-Chief

Paul Brindley

co-Editor-in-Chief
